# Endophytic bacterium *Bacillus aryabhattai* induces novel transcriptomic changes to stimulate plant growth

Hongli Xu[1], Jingyao Gao[1], Roxana Portieles[1], Lihua Du[1], Xiangyou Gao[1], Orlando Borras-Hidalgo[1,2]*

**1** Joint R&D Center of Biotechnology, RETDA, YOTABIO-ENGINEERING CO., LTD., Rizhao, Shandong, P. R. China, **2** State Key Laboratory of Biobased Material and Green Papermaking, Shandong Provincial Key Lab of Microbial Engineering, Qilu University of Technology (Shandong Academy of Science), Jinan, P.R. China

* borrasorlando@yahoo.com

**Data Availability Statement:** Data relevant to this study are available from the BioProject at accession numbers PRJNA781678 and PRJNA781727

## Abstract

In nature, plants interact with a wide range of microorganisms, and most of these microorganisms could induce growth through the activation of important molecular pathways. The current study evaluated whether the endophytic bacterium *Bacillus aryabhattai* encourages plant growth and the transcriptional changes that might be implicated in this effect. The endophytic bacterium promotes the growth of Arabidopsis and tobacco plants. The transcriptional changes in Arabidopsis plants treated with the bacterium were also identified, and the results showed that various genes, such as cinnamyl alcohol dehydrogenase, apyrase, thioredoxin H8, benzaldehyde dehydrogenase, indoleacetaldoxime dehydratase, berberine bridge enzyme-like and gibberellin-regulated protein, were highly expressed. Also, endophytic bacterial genes, such as arginine decarboxylase, D-hydantoinase, ATP synthase gamma chain and 2-hydroxyhexa-2,4-dienoate hydratase, were activated during the interaction. These findings demonstrate that the expression of novel plant growth-related genes is induced by interaction with the endophytic bacterium *B. aryabhattai* and that these changes may promote plant growth in sustainable agriculture.

## Introduction

Plant performance is influenced by the environment and genetic features [1, 2]. Plants are constantly under high pressure from a variety of microbes in their natural environment. Microorganisms and plants establish interesting relationships that are useful for both partners. The cohabitation of microorganisms in a plant exerts an effect on the growth performance of the plant. These interactions effectively enhance the agricultural properties and yields of plants as well as the quality of the soil and nutrient cycling [3–5]. In addition, the extensive application of chemical fertilizer has a negative effect on soil quality and the environment [6]. These negative impacts could be reduced through improved farming practices involving microbial inoculations such as biofertilizers. Bacterial endophytes are used as biofertilizers to enhance crop production and significantly reduce the impact of chemicals in the environment [7–9].

**Funding:** This study was supported by the Special Funds for Guiding Local Science and Technology Development of Central Government of Shandong Province (No. YDZX20193700004362).

**Competing interests:** The authors have declared that no competing interests exist.

Beneficial microorganisms are also used to improve plant yields and constitute sustainable alternatives to chemical fertilizers [10].

Bacillus species comprise the largest class of plant growth-promoting bacteria [11–13]. Bacillus genus members could survive in adverse environments for extended periods of time. Various Bacillus species produce different secondary metabolites capable of inducing plant growth [11]. The use of Bacillus species as biofertilizers provides an alternative for enhancing plant growth and yield [12]. The application of Bacillus species exerts different effects on plants. Most of these effects are related to increases in the length and biomass of shoots, roots, and leaves [13–15]. Bacillus strains can also enhance fruit and grain yields [16, 17]. In addition, biofertilizers involving Bacillus species are more effective at producing diverse metabolites, forming spores, and maintaining cell viability. These characteristics allow the generation of formulated products suitable for commercial use [18].

Endophytic microorganisms produce different bioactive molecules that have a marked direct or indirect effect on plant growth. An understanding of endophytic microorganism-plant interactions may help clarify mechanisms for promoting plant growth and create a sustainable system for crop production [18]. Previous studies have found that *B. methylotrophicus* and *B. subtilis* are involved in hormones synthesis, for instance indole-3-acetic acid, gibberellic acid, 1-aminocyclopropane-1-carboxylate (ACC) deaminase, cytokinins, and spermidines. These proteins are directly involved in the activation of plant growth [15, 19]. Similarly, *B. subtilis* and *B. mojavensis* can secrete ACC deaminase to inhibit plant senescence [20, 21].

Recent studies have shown that inoculation with *Bacillus aryabhattai* impressively improves the nutritional status of wheat crops [22]. The growth and productivity of rice are positively enhanced by treatment with *B. aryabhattai*. This bacterium improves the salt tolerance of plants through increases in atmospheric nitrogen fixation, phosphate solubilization, and indoleacetic acid production [23]. The *B. aryabhattai* strain also promotes the growth of cowpea through increases in indole production, siderophore production, phosphate solubilization, and 1-aminocyclopropane-1-carboxylic acid deaminase activity [24].

*B. aryabhattai* B8W22 can secrete many organic acids, including oxalic acid, malonic acid, citric acid, succinic acid, indole acetic acid (IAA), and siderophores, which can promote plant growth [25]. Most of the studies involving *B. aryabhattai* have focused on the spectrum of plant growth-promoting secondary metabolites produced by this bacterium. However, additional molecular pathways could be involved in both plants and bacteria during this process.

In general, the understanding of the molecular changes established in plants in response to Bacillus species remains limited. Moreover, investigating the genes expressed in Bacillus during the interaction with plants allows a better understanding of the molecular pathways used by bacteria to induce plant growth. The present study aimed to characterize the effects of the endophytic bacteria *B. aryabhattai* based on their plant growth-promoting properties and the main molecular pathways involved in their interaction with Arabidopsis plants.

## Materials and methods

### Endophytic bacterium, plant materials, and growth conditions

The bacterium *Bacillus aryabhattai* was previously isolated and identified from the wild plant species *Glyceria chinensis* (Keng) according to Portieles et al. [26]. Samples were collected from the wild plant species *G. chinensis* (Keng) along the Fu Tuan River (35° 20′ 17″ N, 119° 26′ 8″ E) within 5 km of the coastal region of Rizhao city in Shandong Province, People's Republic of China. *G. chinensis* (Keng) was identified according to data on morphological traits from the Flora of China (http://www.iplant.cn/foc/). This experimental study complied with the national and local laws of China, and sample collection was allowed by the Rizhao

Administration and Municipal Sciences and Technology Department (Collection information: South China Botanical Garden (IBSC) of the Chinese Academy of Sciences. Source: China Digital Plant Specimens Museum. Identifier: 0114164. Collector: Zhang Zhisong Acquisition number: 401467). The strain was cultivated in Luria-Bertani (LB) agar medium (yeast extract, 5 g/l: peptone, 10 g/l; sodium chloride, 5 g/l; agar, 12 g/l; pH 7) at 37 ˚C. Seeds of the *Arabidopsis thaliana* ecotype Columbia were surface-sterilized and placed on Murashige and Skoog (0.5X MS) basal media (Sigma Aldrich, St Louis, MO, USA) supplemented with 1% w/v sucrose. The seeds were maintained in the dark at 4 ˚C for 2 days and transferred to a growth room with a 16-h light/8-h dark photoperiod and a temperature of 22 ˚C. Small plants were sown in a substrate composed of peat plugs and vermiculite (1:1) for 14 days. *Nicotiana tabacum* seeds were also germinated, and plants were cultivated in 6-inch pots with sterilized black turf and rice husk (4:1) substrate and kept at 23 ˚C in a growth room. All the substrates used in this study were sterilized at 120 ˚C for 20 minutes.

Plant material (stems and roots) were first cleaned with water to establish the endophytic property and establishment time of *B. aryabhattai*. The samples were sliced into fragments in an aseptic condition. Each sample was surface sterilized for 1 minute with 70% ethanol before being immersed in a 5% sodium hypochlorite solution for 1 minute. The samples were then washed for 1 minute in sterile distilled water and dried on filter paper. Following adequate drying, plant parts were manually handled for 5 minutes in 1 ml of sterile water in a TissueLyser (Qiagen, Hilden, Germany). The debris was decanted, and 100 l of the remaining water was cultured for three days at 37˚C in Luria-Bertani (LB) agar medium (sodium chloride, 5 g/L; peptone, 10 g/L; yeast extract, 5 g/L; agar, 12 g/L and pH 7). In addition, the final wash solution from the surface sterilization operation was spread out onto an MS medium plate as a control. Internally isolated bacterium was exclusively isolated from processed materials. This was the factor for classifying endophytes as contrast to surface contaminants. Furthermore, the roots of the plants that had been infected with the endophytic bacterium used in the studies was processed, and the bacterium was collected and categorized as the original strain using the same methodology described above. The bacterial colonization in the roots was confirmed using the same method followed in the isolation of this strain on LB plates at various time periods. The endophytic bacterium was fully established 72 hours after inoculation.

## Greenhouse experiments

*Bacillus aryabhattai* was grown for one day at 37 ˚C in the dark in a 250-mL Erlenmeyer flask with 100 mL LB broth medium and 200 rpm shaking. The *B. aryabhattai* fermentation product's optical density (OD) was adjusted to 1.0 (4.77 10$^9$ CFUs/mL), and 30 mL of the fermentation product was poured to each pot. Five *Arabidopsis thaliana* and four *Nicotiana tabacum* plants per plastic pot were cultivated in a growth room at 25 ˚C and maintained with water without fertilizers, respectively. Five-day-old *A. thaliana* and *N. tabacum* plants (seeds previously germinated in MS basal media, transferred to the substrate, and adapted during five days) were treated with the *B. aryabhattai* fermentation product twice weekly for one month. The plant size and the dry and fresh weights were evaluated after one month of treatment with *B. aryabhattai*. Whole plants were used for the evaluation of the different parameters. A completely randomized pot experiment with five replicates of each treatment was performed to analyze the influence of *B. aryabhattai* on the growth of *A. thaliana* and *N. tabacum* plants. The data were analyzed using GraphPad Prism software (La Jolla, CA, USA). A "t" test was used to examine the significance of the differences between the mean values, with P<0.05 indicating a significant difference. Five replicates were subjected to each treatment, and the experiments were replicated three times.

## Identification of novel differentially expressed genes by RNA sequencing

Arabidopsis plants (five-day-old) were treated with 30 mL ($4.77 \times 10^9$ CFUs/mL) of *B. aryabhattai*. Previously, the bacterium was well established 72 h post inoculation according to the protocol mentioned above, and leaves, stems, and roots from five plants were collected after establishment. Plants treated with water were used as a control. The treatments, including the control treatment, were repeated three times per group. In total three libraries were constructed as follow: 1) bacterium; 2) Arabidopsis and 3) Arabidopsis treated with bacterium. The Qiagen RNeasy Midi Kit (Hilden, Germany) was used to extract total RNA, and the concentration of total RNA was measured using spectrometry. Following total RNA extraction and DNase I treatment, magnetic beads containing oligo (dT) were used to isolate mRNA (for eukaryotes) or rRNA (for prokaryotes) using the QIAseq FastSelect 5S/16S/23S Kit (Qiagen, Germany). The samples were sequenced using an Illumina HiSeq™ 2000 (Personalbio Co., Shanghai, People's Republic of China). High-quality reads were processed using the Perl script, and the differentially expressed genes were identified using the edgeR package (https://bioconductor.org/packages/edgeR/) [27]. Genes with a fold change ≥ 2 were considered significantly differentially expressed genes. The differentially expressed genes were characterized using Gene Ontology (GO) and Kyoto Encyclopedia of Genes and Genomes (KEGG) (www.genome.jp/kegg) pathway enrichment analyses [28, 29]. Blast2GO software (www.blast2go.com) was also used to assess the GO annotations [30, 31]. The prokaryote genetic analysis process involved filtering the raw data to obtain a high-quality sequence and comparing the filtered sequences to the reference genome for the species (http://www.tigr.org/softlab, http://sourceforge.net/projects/ngopt) [32, 33].

The read count values of each gene were compared to the gene's original expression level using HTSeq (https://pypi.python.org/pypi/HTSeq) [34]. To make the gene expression levels comparable between different genes and samples, we normalized the sequencing depth and gene length based on fragments per kilobase of transcript per million (FPKM). The FPKM values consider that two reads can map to the same transcript. In the reference transcription group, FPKM > 1 indicated that the gene was expressed. To test the reliability of the experiments and determine whether sample selection was reasonable, Pearson correlation coefficients were calculated to indicate the correlation between the expression levels of genes in the sample and used. The correlation of gene expression levels between samples is an important index indicating the reliability of experiments and whether sample selection was reasonable. The log2-fold change for the up and down regulated plant genes was computed by dividing the FPKM values of Arabidopsis plants treated with the bacterium by the FPKM values of Arabidopsis plants treated with water. While the log2-fold change for the up and down regulated bacterial genes was calculated by dividing the FPKM values of bacterium during the interaction with the plant by the FPKM values of the bacterium grown in LB medium. The 2-fold change cutoff FPKM value was established.

## Analysis of RNA sequencing results

In a second experiment, the transcripts with the largest expression levels from RNA sequencing were confirmed using quantitative real-time polymerase chain reaction. The experiment was done in accordance with the procedure described above. Primer 5.0 was used to generate the oligonucleotides (S1 Table). Total RNA was extracted with a Qiagen RNeasy kit, and cDNA was prepared with oligo-dT primers and a SuperScript III kit (Invitrogen, Carlsbad, CA, USA). A Rotor-Gene Q PCR equipment (Hilden, Germany) and the QuantiTect SYBR Green PCR Kit were used for real-time quantitative PCR (Qiagen). During RT-qPCR gene quantification, the *B. aryabhattai* 16S rRNA and A. thaliana -actin genes were selected as

internal controls. The real-time PCR reaction conditions were as follows: an initial denaturation step at 95 ˚C for 15 minutes, followed by denaturation at 95 ˚C for 15 seconds, an alignment step at 58 ˚C for 30 seconds, and an extension step at 72 ˚C for 40 cycles. Q-Gene software was used to calculate relative gene expression as mean normalized expression [35]. The relative fold-change (log2) values were calculated in relation to the control treatment. All quantitative PCR experiments were replicated three times for biological and technical considerations.

## Results

### *B. aryabhattai* enhances the growth of Arabidopsis and *Nicotiana tabacum* plants

The promotion of the growth of Arabidopsis and *N. tabacum* plants by *B. aryabhattai* was tested in the growth room. Arabidopsis plants treated with *B. aryabhattai* showed increased growth compared with the control plants (Fig 1A). The analysis indicated that bacterial inoculation significantly increased the size of Arabidopsis plants (4.55 cm) compared with that of the control plants (3.10 cm) (Fig 1B). At 20 days posttreatment, bacterial inoculation significantly enhanced the fresh and dry weights of the treated Arabidopsis plants (0.97/0.08 g) compared with those of the control plants (0.47/0.034 g; Fig 1C and 1D). Moreover, *B. aryabhattai* induced the growth of *N. tabacum* plants (Fig 2A). The heights of the inoculated *N. tabacum* plants were significantly greater (4.05 cm) than those of the uninoculated plants (2.25 cm; Fig 2B). A data analysis showed that *N. tabacum* plants inoculated with the endophytic bacterium presented higher fresh and dry weights than the control plants (0.13/0.010 and 0.05/0.003 g, respectively: Fig 2C and 2D). The nutrients from the LB broth medium were a key aspect. In terms of the volume and frequency of LB broth medium applied, this could have a substantial impact factor on the growth of little Arabidopsis plants. However, this had not significant impact on Arabidopsis and tobacco plants. There were not statistically significant changes between the two treatments (S2 Table).

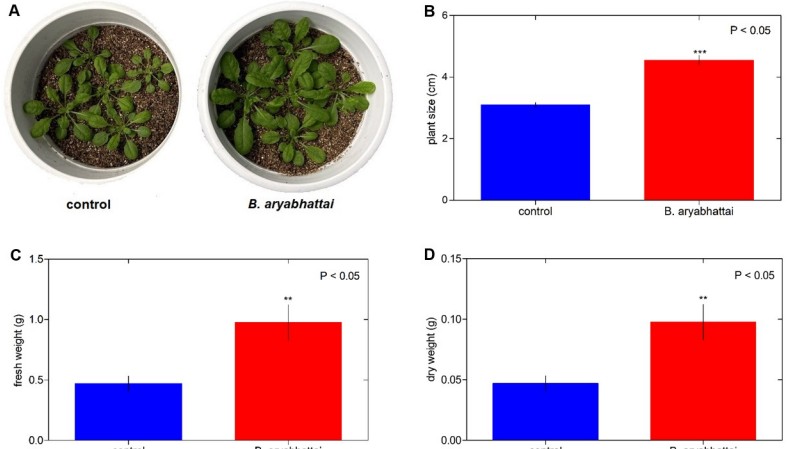

**Fig 1. The endophytic bacterium *B. aryabhattai* enhances Arabidopsis plant growth. A)** Phenotype (at 20 days post inoculation) of mock-treated (control) Arabidopsis plants and those treated with *B. aryabhattai*. Plant size (**B**), fresh (**C**), and dry weight (**D**) of mock-treated and *B. aryabhattai*-treated Arabidopsis plants. The mean results with standard errors from two separate experiments (n = 15) are represented by each bar.

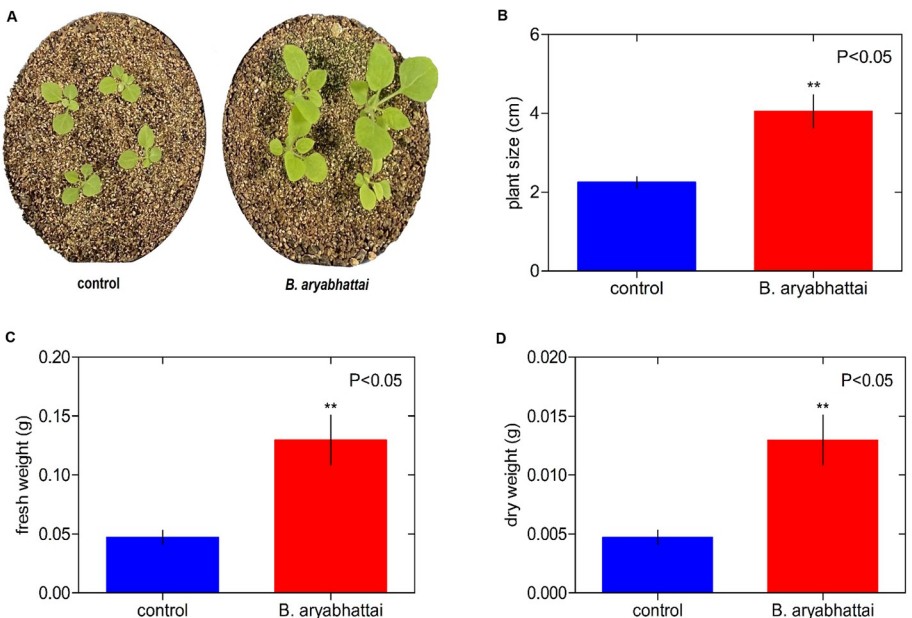

**Fig 2. The endophytic bacterium *B. aryabhattai* enhances *Nicotiana tabacum* plant growth. A)** Phenotype (at 20 days post inoculation) of mock-treated *N. tabacum* plants (control) and those treated with *B. aryabhattai*. Plant size (**B**), fresh (**C**), and dry weight (**D**) of mock-treated and *B. aryabhattai*-treated *N. tabacum* plants. The mean results with standard errors from two separate experiments (n = 15) are represented by each bar.

## *B. aryabhattai* induces transcriptional changes in genes involved in plant growth

Endophytic bacterial species frequently exert important beneficial effects on plant growth. The genes generally associated with the beneficial effects of endophytic bacteria on plant productivity encode proteins involved in different molecular pathways. To determine whether *B. aryabhattai* induces genes associated with the promotion of plant growth, the interactions between *B. aryabhattai* and Arabidopsis were evaluated by RNA-seq. The range of gene expression changes during the *B. aryabhattai*-Arabidopsis interaction was determined. During the interaction, 21,416 transcripts were identified and annotated (S3 Table), and from this set, 6,943 new transcripts were annotated (S4 Table). The highest number of transcripts (33.05%) showed a fold change in expression between 1 and 10, and the lowest number of transcripts exhibited the highest expression level (>100-fold change) (Fig 3).

An analysis of RNA-seq data obtained during the interaction identified 363 differentially expressed transcripts, which included 268 upregulated and 95 downregulated transcripts (Fig 4). Among these transcripts, cinnamyl alcohol dehydrogenase (14983.5-fold), apyrase (2685.5-fold), thioredoxin H8 (2444.5-fold), benzaldehyde dehydrogenase (2166.9-fold), indoleacetaldoxime dehydratase (2092.8-fold), berberine bridge enzyme-like (2018.8-fold), gibberellin-regulated protein (1666.9-fold), maturase K (12.3-fold), tetratricopeptide repeats (TPR)-like superfamily protein (12.0-fold), BTB/POZ, TAZ domain-containing protein (10.8-fold) and auxin-responsive GH3 family protein (10.4-fold) showed the highest differential expression during the *B. aryabhattai*-Arabidopsis interaction. In contrast, zinc finger C-x8-C-x5-C-x3-H type family protein (-293.1-fold), ankyrin repeat/KH domain protein (-119.8-fold), CAPRICE-like MYB3 (-109.6-fold), HSP20-like chaperone superfamily protein (-80.4-fold), gibberellin-regulated protein (-78.8-fold), abscisic acid 8'-hydroxylase (-74.5-fold),

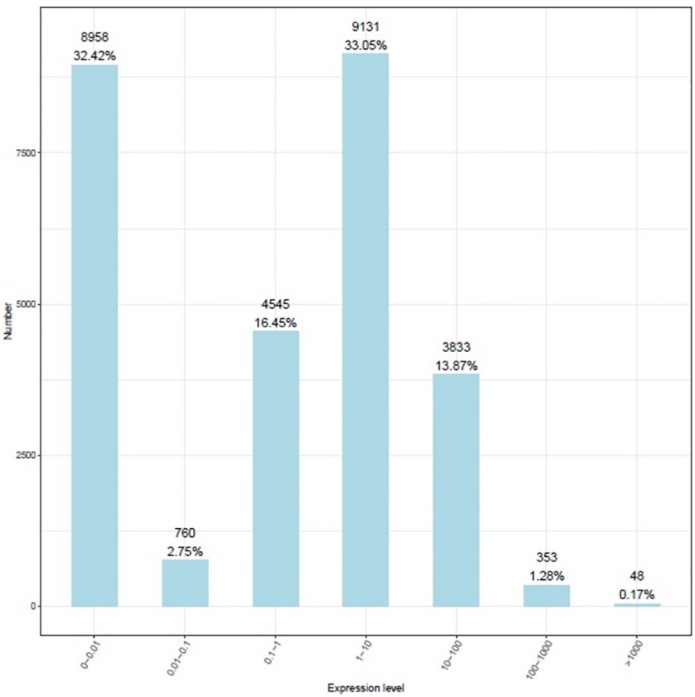

**Fig 3. Plant gene expression during the interaction with *B. aryabhattai*.** The X-axis represents the range of gene expression values, and the Y-axis represents the number of genes in each expression interval.

pectinesterase (-70.1-fold), agamous-like MADS-box protein (-57.2-fold), and ethylene-responsive transcription factor (-51.8-fold) were significantly repressed during the interaction (Table 1).

The differentially expressed transcripts were analyzed to determine their function. A GO enrichment analysis was performed using the annotated differentially expressed genes to generate a list of key genes and the number of genes related to each term. The transcripts showing the largest increases in expression were NADH dehydrogenase (quinone) activity, NADH dehydrogenase (ubiquinone) activity, NADH dehydrogenase activity, and oxidoreductase activity (Fig 5). Based on the KEGG pathways, we categorized the most significant transcripts into the indole alkaloid biosynthesis and linoleic acid metabolism pathways (Fig 6). The expression profile of transcription factor genes revealed the transcription-related activities that are enhanced by these genes. To determine their important roles, the transcription factor genes involved in the *B. aryabhattai*-Arabidopsis interaction were analyzed. The largest transcription factor gene families detected in our study were LOB domain-containing proteins (9.9-fold), zinc finger CCCH domain-containing proteins (6.1-fold), NAC transcription factors (5.6-fold), and BTB and TAZ domain proteins (4.7-fold) (Table 1).

During the interaction, a total of 4,305 *B. aryabhattai* genes were annotated, and the genes from this set that exhibited a log2-fold change in expression between -2.1 and 5.1 were analyzed. Nine and six *B. aryabhattai* genes were up- and downregulated during the interaction with Arabidopsis plants, respectively. The remaining genes had undefined functions, were repeated, had less significant fold changes in expression, or were involved in the primary metabolism of the bacterium. Arginine decarboxylase (5.1-fold), D-hydantoinase (4.9-fold), membrane protein (4.9-fold), ATP synthase gamma chain (4.8-fold), and 2-hydroxyhexa-2,4-dienoate hydratase (4.7-fold) exhibited the highest expression levels in *B. aryabhattai*

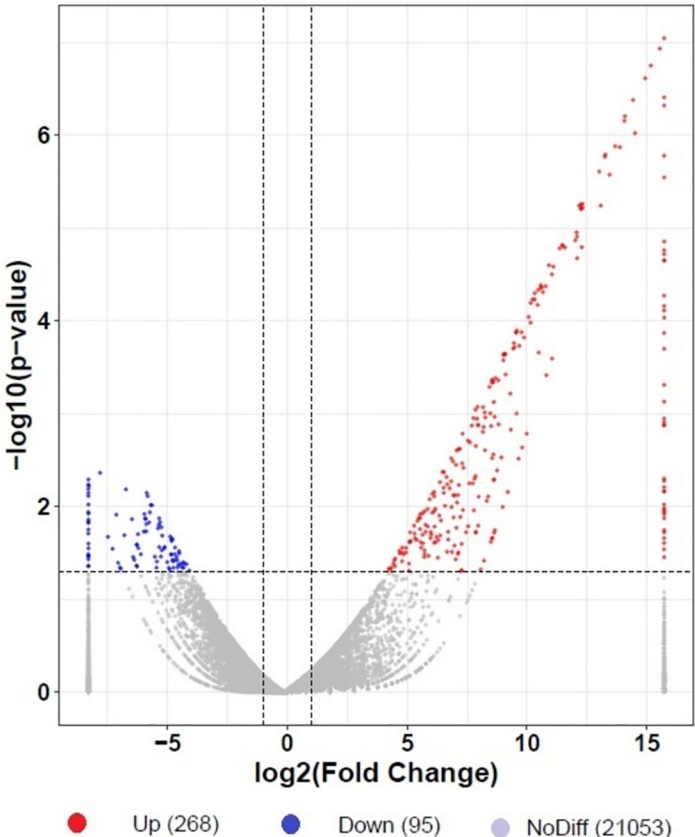

**Fig 4. During the *B. aryabhattai*-Arabidopsis interaction, a volcano plot of differential gene expression was established.** The log2-transformed fold changes are on the X-axis, while the -log10-transformed p values are on the Y-axis. The horizontal dashed lines represent the P value thresholds, while the vertical dashed lines show the difference thresholds (0.05). The scale was set based on the log2-transformed lowest and maximum values (between -7.60 and 15.70).

during the interaction with Arabidopsis plants. Moreover, putative universal stress protein (-3.6-fold), L-lactate dehydrogenase (12.8-fold), succinate dehydrogenase flavoprotein subunit (12.7-fold), and glycolate permease (12.6-fold) were highly downregulated in the bacterium (Table 2 and S5 Table).

## Analysis of RNA sequencing results

Even though RNA-Seq is the fundamental basis for gene expression profiling, qPCR is the main tool for validation. qRT-PCR analysis was done to validate the data generated by RNA sequencing. The expression levels of plant and bacterium genes matched the RNA-Seq data, indicating that the RNA-Seq results are trustworthy (Fig 7). Cinnamyl alcohol dehydrogenase (106-log2 ratio), apyrase (57-log2 ratio) and thioredoxin H8 (35- log2 ratio) were highest expressed genes in Arabidopsis plants inoculated with *B. aryabhattai* in a different replication of the experiment. While the zinc finger C-x8-C-x5-C-x3-H type family protein gene (-27-log2 ratio) was highest repressed during the interaction (Fig 7A). Besides, arginine decarboxylase (8-log2) and D-hydantoinase (6-log2) were highest expressed bacterium genes *in planta*. Meantime, the putative universal stress protein gene (-3-log2) had the most reduced expression (Fig 7B).

**Table 1. Significantly differentially expressed genes during the *B. aryabhattai*-Arabidopsis interaction.**

| ID | Log2(Fold Change) [a] | Description |
|---|---|---|
| **Upregulated genes** | | |
| AT4G37990 | 14983.5 | Cinnamyl alcohol dehydrogenase |
| AT1G14250 | 2685.5 | Apyrase |
| AT1G69880 | 2444.7 | Thioredoxin H8 |
| AT1G04580 | 2166.9 | Benzaldehyde dehydrogenase |
| AT2G30770 | 2092.8 | Indoleacetaldoxime dehydratase |
| AT1G26390 | 2018.8 | Berberine bridge enzyme-like |
| AT3G02885 | 1666.9 | Gibberellin-regulated protein |
| ATCG00040 | 12.3 | Maturase K |
| ATCG00360 | 12.0 | Tetratricopeptide repeat (TPR)-like superfamily protein |
| AT3G48360 | 10.8 | BTB/POZ and TAZ domain-containing protein |
| AT4G37390 | 10.4 | Auxin-responsive GH3 family protein |
| AT4G37540 | 9.9 | LOB domain-containing protein |
| AT4G28040 | 9.7 | WAT1-related protein |
| AT5G09730 | 9.4 | Beta-D-xylosidase |
| AT1G61120 | 8.6 | (E,E)-geranyllinalool synthase |
| AT1G52400 | 7.8 | Beta-D-glucopyranosyl abscisate beta-glucosidase |
| AT4G21680 | 7.8 | Protein NRT1/PTR FAMILY |
| AT4G17470 | 7.7 | Alpha/beta-hydrolase superfamily protein |
| AT1G73220 | 7.5 | Organic cation/carnitine transporter |
| AT3G45140 | 7.3 | Lipoxygenase |
| AT4G15210 | 7.2 | Beta-amylase |
| AT1G54020 | 7.2 | GDSL esterase/lipase |
| ATMG00570 | 7.2 | Sec-independent periplasmic protein translocase |
| AT2G25900 | 6.1 | Zinc finger CCCH domain-containing protein |
| AT1G21310 | 5.8 | Extensin-3 |
| AT2G23170 | 5.7 | Indole-3-acetic acid-amido synthetase |
| AT3G04070 | 5.6 | NAC transcription factor |
| AT5G67480 | 4.8 | BTB and TAZ domain protein |
| AT1G44350 | 4.7 | IAA-amino acid hydrolase |
| AT1G52000 | 4.4 | Jacalin-related lectin |
| **Downregulated genes** | | |
| AT1G29560 | -293.1 | Zinc finger C-x8-C-x5-C-x3-H type family protein |
| AT1G12320 | -119.8 | Ankyrin repeat/KH domain protein |
| AT4G01060 | -109.6 | CAPRICE-like MYB3 |
| AT1G76780 | -80.4 | HSP20-like chaperones superfamily protein |
| AT1G74670 | -78.8 | Gibberellin-regulated protein |
| AT2G29090 | -74.5 | Abscisic acid 8'-hydroxylase |
| AT2G47030 | -70.1 | Pectinesterase |
| AT5G60440 | -57.2 | Agamous-like MADS-box protein |
| AT4G16750 | -51.8 | Ethylene-responsive transcription factor |
| AT5G26749 | -48.5 | C2H2 and C2HC zinc fingers superfamily protein |
| AT2G40330 | -47.5 | Abscisic acid receptor |
| AT2G36270 | -43.7 | Basic-leucine zipper (bZIP) transcription factor |
| AT5G15800 | -42.6 | MADS-box transcription factor family protein |
| AT5G03680 | -33.4 | Trihelix transcription factor |
| AT1G13920 | -32.9 | Remorin family protein |

*(Continued)*

**Table 1.** (Continued）

| ID | Log2(Fold Change) [a] | Description |
|---|---|---|
| AT3G28830 | -31.3 | Mucin-like protein |
| AT4G31370 | -28.6 | Fasciclin-like arabinogalactan protein |
| AT1G16060 | -23.7 | AP2-like ethylene-responsive transcription factor |
| AT1G01380 | -23.2 | MYB-like transcription factor |
| AT5G57640 | -22.1 | GCK domain-containing protein |
| AT4G33970 | -22.1 | Pollen-specific leucine-rich repeat extensin-like protein |
| AT4G31620 | -18.3 | Transcription factor B3 family protein |
| AT5G28640 | -17.8 | GRF1-interacting factor |
| AT3G27650 | -17.2 | LOB domain-containing protein |
| AT4G31380 | -14.0 | Flowering-promoting factor 1-like protein |
| AT5G46530 | -13.4 | AWPM-19-like family protein |
| AT2G45760 | -11.3 | BON1-associated protein |

[a] All the genes with a log2 (fold change) value between -11.3 and 14983.5 were added.

# Discussion

In the current study, we elucidated the mechanism through which *B. aryabhattai* promotes the growth of Arabidopsis and *N. tabacum* plants. *N. tabacum* plants were included to evaluate

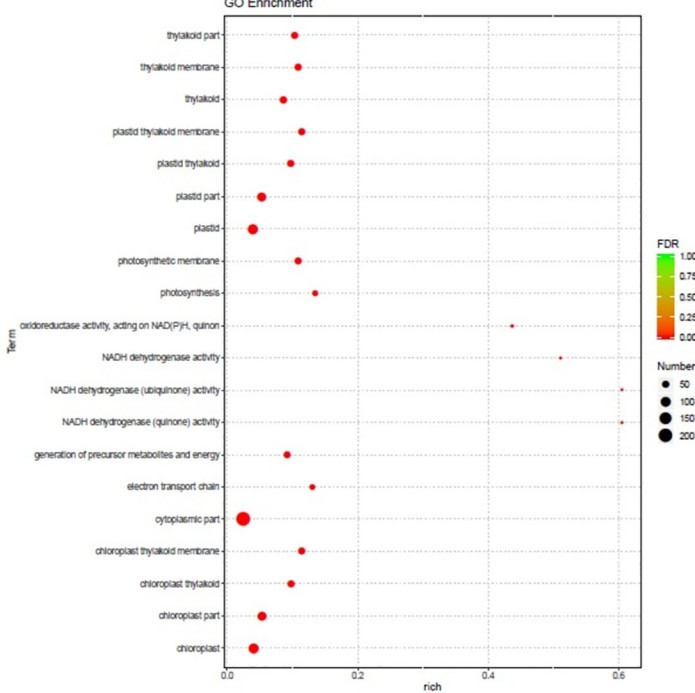

**Fig 5. Gene Ontology (GO) enrichment analysis of the differentially expressed genes during the interaction between *B. aryabhattai* and Arabidopsis.** The rich factor, false discovery rate (FDR) values, and the number of genes with the associated GO term were used to determine the level of enrichment based on the GO enrichment analysis results. The number of distinct genes with the appropriate GO term to the number of annotated genes is referred to as the rich factor. The number of enriched genes is shown by the size of each dot. Each dot is colored differently: red dots represent pathways with high concentration, whereas green dots suggest pathways with low concentration.

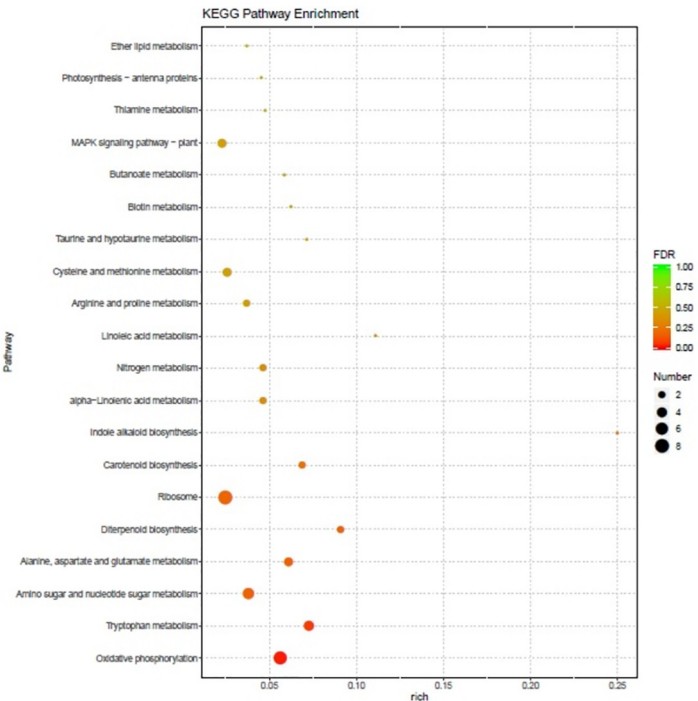

**Fig 6. KEGG pathway analysis of differentially expressed genes during the *B. aryabhattai*-Arabidopsis interaction.** The chart shows the enrichment of signaling pathways with differentially expressed genes. The Y-axis label represents the pathway, and the X-axis label represents the rich factor (rich factor = number of differentially expressed genes enriched in the pathway/number of genes in the background gene set). The size and color of the bubble represent the number of differentially expressed genes enriched in the pathway and the enrichment significance, respectively.

**Table 2. Highly differentials expressed genes in *B. aryabhattai* during its interaction with Arabidopsis plants.**

| ID | Log2(Fold Change) [a] | Swiss-Prot |
|---|---|---|
| **Upregulated genes** | | |
| gene3107 | 5.1 | Arginine decarboxylase |
| gene1769 | 4.9 | D-hydantoinase |
| gene1522 | 4.9 | Membrane protein |
| gene4310 | 4.8 | ATP synthase gamma chain |
| gene1091 | 4.7 | 2-hydroxyhexa-2,4-dienoate hydratase |
| gene5616 | 3.2 | Transcriptional regulatory protein |
| gene3481 | 2.6 | Microcystinase C |
| gene2400 | 2.4 | 4,4'-diapolycopen-4-al dehydrogenase |
| gene1987 | 2.1 | Succinate-CoA ligase [ADP-forming] subunit beta |
| **Downregulated genes** | | |
| gene4030 | -3.6 | Putative universal stress protein |
| gene5157 | -2.8 | L-lactate dehydrogenase |
| gene1475 | -2.7 | Succinate dehydrogenase flavoprotein subunit |
| gene5156 | -2.6 | Glycolate permease |
| gene4267 | -2.1 | Betaine aldehyde dehydrogenase |
| gene4268 | -2.1 | Alcohol dehydrogenase |

[a] All the genes with a log2 (fold change) value between -2.1 and 5.1 were added.

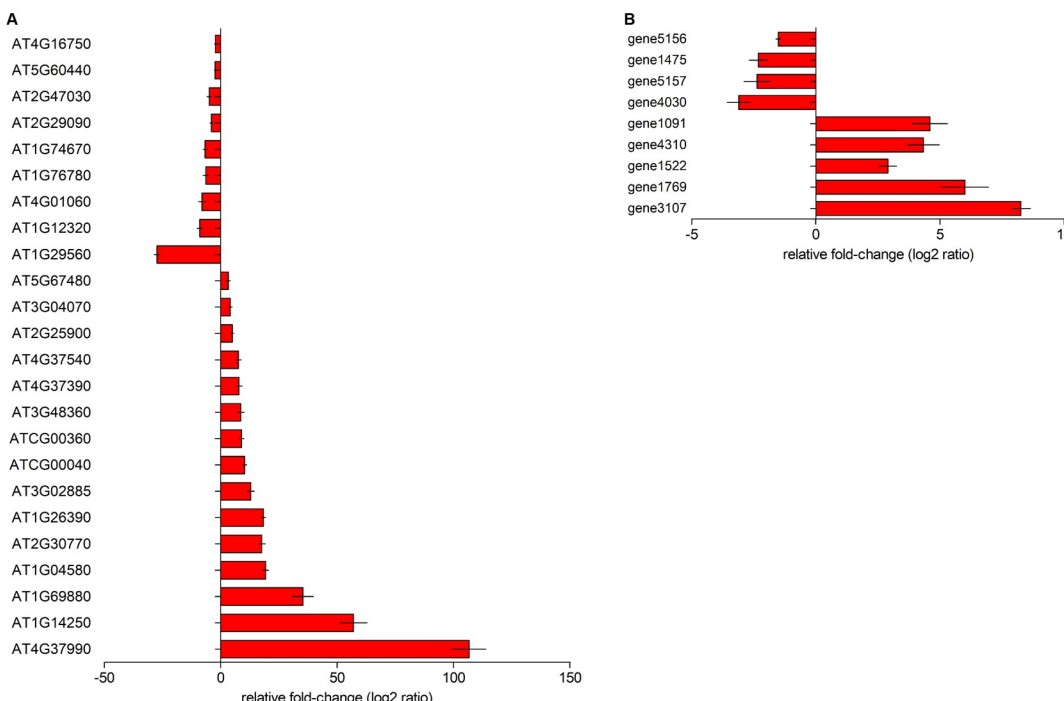

**Fig 7. Analysis of differential expressed genes using qRT-PCR. A)** Expression levels of Arabidopsis plants genes during the interaction with *B. aryabhattai*. **B)** Expression of bacterium genes *in planta*.

their influence on other plant species. The bacterium increases the plant size and fresh and dry weights. Notably, although this endophytic bacterium was isolated from wild plant species, it exerts a robust impact on the growth of nonhost plants. The findings indicate that *B. aryabhattai* can stimulate growth in a range of plant species. However, these effects should be studied in a natural context and with other crop species. Regardless of the context, based on the effect of *B. aryabhattai* on plants, this bacterium could be used as a biofertilizer to boost growth and agricultural production.

The beneficial effect of different Bacillus species on various plant species was previously established [11, 13, 15]. Bacillus species are the most common type of growth-promoting bacterium, and these bacterial usually promote growth via growth-responsive genes, proteins, phytohormones, and metabolites [11]. Additionally, Bacillus is reportedly effective in increasing the biomass and height of important crops through increases in plant N uptake, phosphate solubilization and root-promoting phytohormones [36–38]. Additionally, most of these effects are associated with the production of phytohormones such as indole-3-acetic acid (IAA), cytokinins, gibberellic acid (GA), and spermidines [9, 15, 19]. Furthermore, the induction of endogenous proteins, amino acids, and minerals by Bacillus species could promote plant growth [15, 39].

Beneficial plant-bacteria interactions have been extensively analyzed. However, the specific molecular pathways that are associated with these interactions are unclear. This knowledge is critical for maximizing the potential of these microbes in the field. In the current study, we done RNA sequencing to analyze the genes expressed during the Arabidopsis-*B. aryabhattai* interaction. We conducted an analysis of the RNA-seq dataset and found various types of genes that might be ideal candidates for characterizing the host pathway governing this essential plant-microbe relationship. A high number of novel genes involved in metabolite

biosynthesis were differentially expressed in our dataset. The results reveal new insights into plant and bacterial gene expression and improve our understanding of the molecular events involved in the Arabidopsis-*B. aryabhattai* interaction. Notably, GO and KEGG analyses revealed significant changes between treated and nontreated plants. Our data indicate that *B. aryabhattai* triggers important molecular pathways related to plant growth.

Interestingly, the KEGG pathway analysis revealed that the most significant transcripts were involved in the indole alkaloid biosynthesis and linoleic acid metabolism pathways. These pathways have been related to resistance and tolerance to biotic and abiotic factors [40, 41]. Most likely, the bacterium can induce plant protection against pathogens and abiotic factors through the production of jasmonic acid and secondary metabolites associated with the defense responses of plants. The synthesis of indole-alkaloids in plants causes consistent alterations of the microbiota surviving at the barley root-soil interface while having no discernible detrimental influence on plant growth performance in two elite barley types [42]. These results indicate that the application of indole-alkaloids modulates the proliferation of a subset of soil microbes with relatively broad phylogenetic assignments [42].

In contrast, bacteria can undergo symbiotic or pathogenic interactions with plants. Membrane lipids and lipid-derived molecules from the plant or microbial organism play important roles during the infection process [43]. For example, lipids are involved in establishing the membrane interface between the two organisms [43]. Furthermore, lipid-derived molecules are crucial for intracellular signaling in plant cells, and lipids serve as signals during plant-microbe communication [43]. Linolenic acid is released from several complex fatty acids mainly located in the membranes of organelles such as chloroplasts and is a precursor of jasmonic acid [44, 45]. The involvement of this hormone in different biological processes of plants, such as responses to biotic and abiotic stress conditions, indicates the possible use of this bacterium to activate plant defense.

Surprisingly, during the Arabidopsis-*B. aryabhattai* interaction, the expression of genes encoding cinnamyl alcohol dehydrogenase, apyrase, thioredoxin H8, benzaldehyde dehydrogenase, indoleacetaldoxime dehydratase, berberine bridge enzyme-like, gibberellin-regulated protein, maturase K, tetratricopeptide repeat (TPR)-like superfamily protein, BTB/POZ and TAZ domain-containing protein and auxin-responsive GH3 family protein genes were highly induced. Some identified genes from our dataset are addressed here, as well as their roles in other plant-microbe relationships. *B. aryabhattai* most likely promotes the synthesis of lignin during the growing phase of Arabidopsis and *N. tabacum* plants, which results in the robust phenotype observed in the plants treated with the bacterium. Cinnamyl alcohol dehydrogenase is a key enzyme involved in lignin synthesis and is closely related to plant growth and development. This enzyme is expressed in the lateral roots and root tips of sweet potato, and its activity is induced by abscisic acid [46].

Similarly, apyrases play a crucial role in regulating the growth of Arabidopsis plants treated with the bacterium. Specifically, apyrases influence auxin transport and stomatal aperture, and the removal of apyrase activity can lead to growth inhibition [47]. A previous study revealed that silencing of the apyrase gene induces significant phenotypic changes, growth retardation, an increase in the tuber number per plant, and an effect on the tuber morphology of potato plants [48]. Moreover, the expression of apyrase genes in Arabidopsis plants exerts a marked effect on the growth of plant tissues and the accumulation of auxin [49, 50].

Interestingly, *B. aryabhattai* can indirectly induce plant growth through benzaldehyde dehydrogenase, which plays a key role in the benzoic acid pathway. In addition, benzaldehyde dehydrogenase is involved in the processing of benzaldehyde to benzoic acid. The growth, mineral composition, and chlorophyll content of soybean plants are influenced by benzoic

acid [51]. Herein, benzoic acid exerts a marked effect on the growth and yield of tomato plants. Similarly, benzoic acid exerts a positive effect on fruit yield [52].

We observed high induction of maturase K gene expression during the interaction. Recent studies revealed that the maturase K gene is induced during the Arabidopsis-*Bacillus altitudinis* interaction [53]. In addition, the maturase K gene is highly expressed in *Anoectochilus roxburghii* plants treated with endophytic fungi [54]. Most likely, this gene could be related to the Bacillus-plant association, but a functional analysis of Arabidopsis mutants could provide a clear understanding of the real function of this gene during Bacillus-plant interactions.

Furthermore, tetratricopeptide repeat (TPR)-like superfamily protein genes might have indirect effects on the activation of phytohormones related to plant growth. This gene was found to be expressed during the interaction and constitutes a basic component of gibberellin and ethylene responses. The silencing of an Arabidopsis chloroplast-localized tetratricopeptide repeat protein gene affects plant growth, leaf greening, chloroplasts, and photosynthesis genes [55].

We also predicted that hormone-related genes and transcription factors would be modulated during this interaction. The bacterium *B. aryabhattai* induced plant growth by triggering key molecular pathways involved in the production of phytohormones and transcription factors. Root development, shoot growth, and fruit ripening are regulated by Aux/IAA family genes [56]. Auxin influences numerous stages of plant development and growth by regulating the expression of auxin-activated genes [57], and auxin controls plant development and growth by altering the expression of different genes [58]. This finding could be directly correlated with the phenotype observed in the Arabidopsis plants treated with the bacteria.

Many of the processes that occur in a plant during interactions with endophytic bacteria are known. It is also important to understand the processes that take place in a bacterium during its interaction with a plant, such as the specific genes that are expressed in the bacterium that may contribute to the growth phenotype of treated plants. This point is more complex, as demonstrated by the low proportion of bacterial transcripts found during the interaction. A functional analysis of these types of genes would elucidate the actual role of these genes during the interaction. Interestingly, the expression of arginine decarboxylase, D-hydantoinase, ATP synthase gamma chain and 2-hydroxyhexa-2,4-dienoate hydratase genes was highly induced in *B. aryabhattai* during its interaction with the plant, and this finding constitutes the first line of evidence indicating that these types of genes are expressed in this species. We speculate that the overexpression of these genes in *B. aryabhattai* might enhance plant growth. Based on the same principles, we discuss some bacterial genes expressed during the interaction.

For example, the activity of arginine decarboxylase has been implicated in the effect of hormones on plant growth [59]. The expression of this enzyme is correlated with cell growth and stress responses in apple plants [60]. Additionally, this enzyme is involved in efficient ROS elimination and its influence on root growth, which is conducive to drought tolerance [61]. Most of the effects of these genes could be indirectly implicated in enhancement of the plant physiological status and thus in a better growing environment.

Contradictorily, the expression of the D-hydantoinase gene was induced in the bacterium during its interaction with the plant. These D-amino acids (D-AAs) exert a growth-inhibiting effect on plants [62–64]. The exogenous administration of various D-AAs to growth media has a deleterious or favorable effect on the growth and development of many plant species, and these effects depend on the D-AAs used [63, 65]. However, the mechanism responsible for the finding that its expression is correlated with growth stimulation but not growth inhibition. Although some D-AAs impede seedling growth, some lines of evidence show that D-AAs can also promote plant growth. Recent studies found that the application of 0.1 mM d-Leu, d-Val, and d-Cys increases the growth of pepper plants [66]. Additionally, Arabidopsis seedlings treated with d-Lys and d-Ile at a range of 1–10 mM exhibit superior growth. This ostensible

inconsistency has been resolved through functional studies. Another conclusion is that D-AAs should not be considered a class of chemicals with uniform features but rather that each D-AA should be evaluated separately.

Among the identified group of plant genes and those showing the most differential expression, the zinc finger C-x8-C-x5-C-x3-H type family protein, ankyrin repeat/KH domain protein, CAPRICE-like MYB3, HSP20-like chaperone superfamily protein and gibberellin-regulated protein were downregulated during the interaction. Additionally, bacterial genes such as putative universal stress protein, L-lactate dehydrogenase, and succinate dehydrogenase flavoprotein subunit showed the strongest repression during the interaction.

For example, the zinc finger C-x8-C-x5-C-x3-H type family protein functions as a transcriptional activator and is involved in secondary wall biosynthesis. Additionally, some evidence shows that this type of protein is involved in ABA-, GA- and phytochrome-mediated seed germination responses [67]. Furthermore, the ankyrin repeat protein family plays a crucial role in plant growth and development and in the responses to biotic and abiotic stresses [68, 69]. In addition, CAPRICE-like MYB3 encodes a small protein with an R3 MYB motif and promotes root hair cell differentiation in Arabidopsis plants. The overexpression of this protein results in the suppression of trichomes and the overproduction of root hairs and exerts pleiotropic effects on flowering development, epidermal cell size and trichome branching [70, 71]. Similarly, HSP20-like chaperone superfamily proteins mediate protein folding and are associated with abiotic stresses and death [72]. Gibberellin-regulated proteins are important endogenous plant growth regulators involved in different physiological processes, and these types of proteins are sometimes up- or downregulated depending on the developmental stages of the plants [73]. Recent studies have shown that orange gibberellin-regulated proteins are involved in allergy reactions [74].

Among the downregulated bacterial genes, putative universal stress proteins are important elements for survival under anaerobic conditions and during persistent colonization and infection with pathogenic bacteria [75]. In addition, lactate dehydrogenase plays an important role in the anaerobic metabolic pathway and catalyzes the reversible conversion of lactate to pyruvate with the reduction of NAD+ to NADH [76, 77]. Interestingly, the succinate dehydrogenase flavoprotein subunit is involved in cellular energetics and is needed for the virulence of many important bacterial pathogens [78]. Certainly, most of these genes could play an important role during the interaction, but functional analyses are needed to draw a conclusion.

## Conclusions

In summary, our results show that the endophytic bacterium *B. aryabhattai* significantly promotes the growth of Arabidopsis and *N. tabacum* plants. Notably, the expression of novel genes involved in different plant growth pathways is involved identified during the Arabidopsis-*B. aryabhattai* interaction. In contrast, the bacterial genes expressed during the interaction could produce some proteins, enzymes, and secondary metabolites capable of inducing plant growth-promoting genes, but further investigation of this finding is needed. Similarly, the effect of this bacterium on important crops and field conditions needs to be evaluated. The potential beneficial effects of *B. aryabhattai* identified in this study suggest that this bacterium is an appropriate and efficient candidate for use in sustainable agriculture.

## Supporting information

**S1 Table. List of oligonucleotides used for RNA sequencing validation.**
(XLSX)

**S2 Table. Evaluation of the effect of LB broth medium on Arabidopsis and tobacco growth experiment.**
(XLSX)

**S3 Table. Plant gene annotation during Arabidopsis-*Bacillus aryabhattai* interaction.**
(XLSX)

**S4 Table. Plant new gene annotation during Arabidopsis-*Bacillus aryabhattai* interaction.**
(XLSX)

**S5 Table. *Bacillus aryabhattai* gene annotation during the interaction.**
(XLSX)

## Author Contributions

**Conceptualization:** Roxana Portieles, Orlando Borras-Hidalgo.

**Data curation:** Hongli Xu, Jingyao Gao, Roxana Portieles, Orlando Borras-Hidalgo.

**Formal analysis:** Jingyao Gao, Roxana Portieles, Orlando Borras-Hidalgo.

**Funding acquisition:** Lihua Du, Xiangyou Gao, Orlando Borras-Hidalgo.

**Investigation:** Hongli Xu, Jingyao Gao, Roxana Portieles, Lihua Du, Xiangyou Gao, Orlando Borras-Hidalgo.

**Methodology:** Hongli Xu, Jingyao Gao, Roxana Portieles, Lihua Du, Orlando Borras-Hidalgo.

**Project administration:** Hongli Xu, Lihua Du, Xiangyou Gao, Orlando Borras-Hidalgo.

**Resources:** Lihua Du, Orlando Borras-Hidalgo.

**Software:** Orlando Borras-Hidalgo.

**Supervision:** Hongli Xu, Xiangyou Gao, Orlando Borras-Hidalgo.

**Validation:** Hongli Xu, Jingyao Gao, Roxana Portieles, Orlando Borras-Hidalgo.

**Visualization:** Lihua Du, Xiangyou Gao, Orlando Borras-Hidalgo.

**Writing – original draft:** Orlando Borras-Hidalgo.

**Writing – review & editing:** Orlando Borras-Hidalgo.

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
