## [Decision Letter · Decision Letter 0]

27 May 2022

PONE-D-22-09489Endophytic bacterium Bacillus aryabhattai induces novel transcriptomic changes to stimulate plant growth.PLOS ONE

Dear Dr. Borras-Hidalgo,

Thank you for submitting your manuscript to PLOS ONE. After careful consideration, we feel that it has merit but does not fully meet PLOS ONE’s publication criteria as it currently stands. Therefore, we invite you to submit a revised version of the manuscript that addresses the major points raised during the review process.

Especially,Comments 1, 2, 3, and 4 from reviewers #1. It is important to give analysis and conclusion of the RNAseq data.As suggested from reviewer #2, qRT-PCR assay is required  to prove the RNA seq data. This can be focused on keys genes from your RNA seq result. 

If you can fulfil the above requirement, please submit your revised manuscript by Jul 11 2022 11:59PM. If you will need more time than this to complete your revisions, please reply to this message or contact the journal office at plosone@plos.org. Please include the following items when submitting your revised manuscript:A rebuttal letter that responds to each point raised by the academic editor and reviewer(s). You should upload this letter as a separate file labeled 'Response to Reviewers'.A marked-up copy of your manuscript that highlights changes made to the original version. You should upload this as a separate file labeled 'Revised Manuscript with Track Changes'.An unmarked version of your revised paper without tracked changes. You should upload this as a separate file labeled 'Manuscript'.

We look forward to receiving your revised manuscript.

Kind regards,

Ching-Hong Yang

Academic Editor

PLOS ONE

[This study was supported by the Special Funds for Guiding Local Science and Technology Development of Central Government of Shandong Province (No. YDZX20193700004362).]

 [This study was supported by the Special Funds for Guiding Local Science and Technology Development of Central Government of Shandong Province (No. YDZX20193700004362).]

[The authors have declared that no competing interests exist.] 

We note that one or more of the authors are employed by a commercial company: RETDA, YOTABIO-ENGINEERING CO., LTD.

4. Please upload a copy of Supporting Information Tables S1, S2 and S3 which you refer to in your text on pages 5 and 6.

5.  Thank you for submitting the above manuscript to PLOS ONE. During our internal evaluation of the manuscript, we found significant text overlap between your submission and the following previously published works, some of which you are an author.

- https://www.frontiersin.org/articles/10.3389/fmicb.2021.692313/full

- https://www.nature.com/articles/s41598-021-91837-5

- https://peerj.com/articles/12498/

- https://facultyopinions.com/prime/ext/726183569

Please revise the manuscript to rephrase the duplicated text, cite your sources, and provide details as to how the current manuscript advances on previous work. Please note that further consideration is dependent on the submission of a manuscript that addresses these concerns about the overlap in text with published work.

Reviewers' comments:

Reviewer's Responses to Questions

**Comments to the Author**

1. Is the manuscript technically sound, and do the data support the conclusions?

Reviewer #1: Partly

Reviewer #2: Partly

2. Has the statistical analysis been performed appropriately and rigorously? 

Reviewer #1: N/A

Reviewer #2: Yes

3. Have the authors made all data underlying the findings in their manuscript fully available?

Reviewer #1: Yes

Reviewer #2: Yes

4. Is the manuscript presented in an intelligible fashion and written in standard English?

Reviewer #1: No

Reviewer #2: Yes

5. Review Comments to the Author

Reviewer #1: This manuscript addresses the Endophytic bacterium Bacillus aryabhattai promotes the growth of Arabidopsis thaliana and Nicotiana tabacum. Although the great potential of various Bacillus spp. in plant growth promotion has been demonstrated in several important crop plants, the molecular mechanism underlying the gene regulation is still unclear. In this study, the RNAseq assay was performed and differentially expressed genes (DEGs) were analyzed in the plant as well as Bacillus aryabhattai. RNAseq data might provide a new highlight into the plant growth promotion under the mutual reaction between the plant and bacteria. Unfortunately, the manuscript has flaws in experiments design, data collection, and interpretation that will prevent publication in its current form.

Major concerns:

1. 30 ml of LB culture with Bacillus aryabhattai was applied to Arabidopsis seedlings twice a week. However, “Plants treated with water were used as a control.” Based on this experiment design, the nutrient from LB broth was a significant impact factor for the growth of tiny Arabidopsis seedlings in terms of the volume and frequency of LB applied.

2. What is the method for RNA extraction from Bacillus aryabhattai? What are the control and treatment of Bacillus aryabhattai?

3. Please clarify the cutoff RPKM value. Could you explain the data from Table 1 “AT4G37990, log2(Fold change) = 14983.5”?

4. In the results part, there is no analysis and conclusion for the RNAseq data. It is not helpful for readers to understand the meanings of RNAseq data by simply listing gene, GO, and KEGG names.

5. Information is not consistent. For example,

A. “Three plants were grown per plastic pot” was claimed but five plants were shown in Figure 1A.

B. “Small plants were sown in a substrate composed of peat plugs and vermiculite (1:1) for 14 days.” “Five-day-old Arabidopsis thaliana and Nicotiana tabacum plants were treated with the B. aryabhattai fermentation product twice weekly for one month.” “At 20 days posttreatment, bacterial inoculation significantly enhanced the fresh and dry weights of the treated Arabidopsis plants.” It is very confused about the days from these statements. How did the author define Five-day-old Arabidopsis thaliana? The seeds were germinated on the 1/2MS plates and transferred into pots. When did the author start to count days?

6. “The bacterium was well established 72 h post inoculation, and leaves, stems, and roots from five plants were collected after establishment.”

A. Is there any evidence to support the statement “The bacterium was well established 72 h post inoculation”?

B. Does this statement imply tissues were collected after 72 h?

Reviewer #2: The MS showed that the endophytic bacterium promotes the growth of Arabidopsis and tobacco plants. The RNA seq data showed that various genes were highly expressed in Arabidopsis plants treated with the bacterium and also some genes of the endophytic bacterial were activated during the interaction. However, no qPCR to prove the RNA seq data.

6. PLOS authors have the option to publish the peer review history of their article (what does this mean?). If published, this will include your full peer review and any attached files.

Reviewer #1: No

Reviewer #2: No

---

## [Author Response · Author response to Decision Letter 0]

14 Jun 2022

Dear Prof. Dr. Ching-Hong Yang

Academic Editor

PLOS ONE

This study was supported by the Special Funds for Guiding Local Science and Technology Development of Central Government of Shandong Province (No. YDZX20193700004362). The funders had no role in study design, data collection and analysis, decision to publish, or preparation of the manuscript.

With this revision we would like to submit a revised version of our manuscript entitled "Endophytic bacterium Bacillus aryabhattai induces novel transcriptomic changes to stimulate plant growth". We would like to thank you and the referees for the excellent comments made to our manuscript and we feel the comments helped to prepare an improved version of our manuscript.

Here, we would like to address all comments made by the reviewers and the changes we have made to meet those comments. A revised version of the manuscript was submitted in which all changes are indicated in “Red Font” to facilitate the revision procedure.

Best regards

Prof. Orlando Borras-Hidalgo

Reviewers' comments:

Reviewer #1: This manuscript addresses the Endophytic bacterium Bacillus aryabhattai promotes the growth of Arabidopsis thaliana and Nicotiana tabacum. Although the great potential of various Bacillus spp. in plant growth promotion has been demonstrated in several important crop plants, the molecular mechanism underlying the gene regulation is still unclear. In this study, the RNAseq assay was performed and differentially expressed genes (DEGs) were analyzed in the plant as well as Bacillus aryabhattai. RNAseq data might provide a new highlight into the plant growth promotion under the mutual reaction between the plant and bacteria. Unfortunately, the manuscript has flaws in experiments design, data collection, and interpretation that will prevent publication in its current form.

Major concerns:

1. 30 ml of LB culture with Bacillus aryabhattai was applied to Arabidopsis seedlings twice a week. However, “Plants treated with water were used as a control.” Based on this experiment design, the nutrient from LB broth was a significant impact factor for the growth of tiny Arabidopsis seedlings in terms of the volume and frequency of LB applied.

Authors: Excellent observation. We agreed with the reviewer. In the previous version of our manuscript this analysis previously done was not included. We have included this point requested by the reviewer as supporting information in the current version. Basically, we also evaluated the effect of LB medium on Arabidopsis and tobacco plants. We did not find significant differences between the treatment with water and LB medium. Taking account, the reviewer remark, we have included this in the “Results” section as follow:

“The nutrients from the LB broth medium were a key aspect. In terms of the volume and frequency of LB broth medium applied, this could have a substantial impact factor on the growth of little Arabidopsis plants. However, this had not significant impact on Arabidopsis and tobacco plants. There were not statistically significant changes between the two treatments (S2 Table).”

2. What is the method for RNA extraction from Bacillus aryabhattai? What are the control and treatment of Bacillus aryabhattai?

Authors: We agree with the reviewer. This is not clarified in the manuscript. We have included this information in “Materials and Methods” section.

“In total three libraries were constructed as follow: 1) bacterium; 2) Arabidopsis and 3) Arabidopsis treated with bacterium. The Qiagen RNeasy Midi Kit (Hilden, Germany) was used to extract total RNA, and the concentration of total RNA was measured using spectrometry. Following total RNA extraction and DNase I treatment, magnetic beads containing oligo (dT) were used to isolate mRNA (for eukaryotes) or rRNA (for prokaryotes) using the QIAseq FastSelect 5S/16S/23S Kit (Qiagen, Germany).”

3. Please clarify the cutoff FPKM value. Could you explain the data from Table 1 “AT4G37990, log2(Fold change) = 14983.5”?

Authors: We agree with the reviewer. This is not clear in the manuscript. This remark was added as follow in the “Materials and Methods” section.

“The log2-fold change for the up and down regulated plant genes was computed by dividing the FPKM values of Arabidopsis plants treated with the bacterium by the FPKM values of Arabidopsis plants treated with water. While the log2-fold change for the up and down regulated bacterial genes was calculated by dividing the FPKM values of bacterium during the interaction with the plant by the FPKM values of the bacterium grown in LB medium. The 2-fold change cutoff FPKM value was established.”

4. In the results part, there is no analysis and conclusion for the RNAseq data. It is not helpful for readers to understand the meanings of RNAseq data by simply listing gene, GO, and KEGG names.

Authors: We agree with the reviewer and a remark based on his comment has been added in the “Results” section.

“Even though RNA-Seq is the fundamental basis for gene expression profiling, qPCR is the main tool for validation. qRT-PCR analysis was done to validate the data generated by RNA sequencing. The expression levels of plant and bacterium genes matched the RNA-Seq data, indicating that the RNA-Seq results are trustworthy (Fig 7). Cinnamyl alcohol dehydrogenase (106-log2 ratio), apyrase (57-log2 ratio) and thioredoxin H8 (35- log2 ratio) were highest expressed genes in Arabidopsis plants inoculated with B. aryabhattai in a different replication of the experiment. While the zinc finger C-x8-C-x5-C-x3-H type family protein gene (-27-log2 ratio) was highest repressed during the interaction (Fig 7A). Besides, arginine decarboxylase (8-log2) and D-hydantoinase (6-log2) were highest expressed bacterium genes in planta. Meantime, the putative universal stress protein gene (-3-log2) had the most reduced expression (Fig 7B).”

5. Information is not consistent. For example,

A. “Three plants were grown per plastic pot” was claimed but five plants were shown in Figure 1A.

Authors: We agree with the reviewer. It was a mistake. This point was clarified in the document as follow in the “Materials and Methods” section:

“Five Arabidopsis thaliana and four Nicotiana tabacum plants per plastic pot were cultivated in a growth room at 25 °C and maintained with water without fertilizers, respectively.”

B. “Small plants were sown in a substrate composed of peat plugs and vermiculite (1:1) for 14 days.” “Five-day-old Arabidopsis thaliana and Nicotiana tabacum plants were treated with the B. aryabhattai fermentation product twice weekly for one month.” “At 20 days posttreatment, bacterial inoculation significantly enhanced the fresh and dry weights of the treated Arabidopsis plants.” It is very confused about the days from these statements. How did the author define Five-day-old Arabidopsis thaliana? The seeds were germinated on the 1/2MS plates and transferred into pots. When did the author start to count days?

Authors: We agree with the reviewer. This point was clarified in the document as follow in the “Materials and Methods” section:

“Five-day-old A. thaliana and N. tabacum plants (seeds previously germinated in MS basal media, transferred to the substrate, and adapted during five days) were treated with the B. aryabhattai fermentation product twice weekly for one month.”

6. “The bacterium was well established 72 h post inoculation, and leaves, stems, and roots from five plants were collected after establishment.”

A. Is there any evidence to support the statement “The bacterium was well established 72 h post inoculation”?

Authors: We agree with the reviewer. We have clarified this point in the manuscript in the “Material and Methods” section as follow:

“Plant material (stems and roots) were first cleaned with water to establish the endophytic property and establishment time of B. aryabhattai. The samples were sliced into fragments in an aseptic condition. Each sample was surface sterilized for 1 minute with 70% ethanol before being immersed in a 5% sodium hypochlorite solution for 1 minute. The samples were then washed for 1 minute in sterile distilled water and dried on filter paper. Following adequate drying, plant parts were manually handled for 5 minutes in 1 ml of sterile water in a TissueLyser (Qiagen, Hilden, Germany). The debris was decanted, and 100 l of the remaining water was cultured for three days at 37°C in Luria-Bertani (LB) agar medium (sodium chloride, 5 g/L; peptone, 10 g/L; yeast extract, 5 g/L; agar, 12 g/L and pH 7). In addition, the final wash solution from the surface sterilization operation was spread out onto an MS medium plate as a control. Internally isolated bacterium was exclusively isolated from processed materials. This was the factor for classifying endophytes as contrast to surface contaminants. Furthermore, the roots of the plants that had been infected with the endophytic bacterium used in the studies was processed, and the bacterium was collected and categorized as the original strain using the same methodology described above. The bacterial colonization in the roots was confirmed using the same method followed in the isolation of this strain on LB plates at various time periods. The endophytic bacterium was fully established 72 hours after inoculation.” 

B. Does this statement imply tissues were collected after 72 h?

Authors: Yes, considering that we already knew when the bacterium was well-established in the interior tissue. 

Reviewer #2: The MS showed that the endophytic bacterium promotes the growth of Arabidopsis and tobacco plants. The RNA seq data showed that various genes were highly expressed in Arabidopsis plants treated with the bacterium and also some genes of the endophytic bacterial were activated during the interaction. However, no qPCR to prove the RNA seq data.

Authors: We agree with the reviewer. Now, we have included this information in detail in “Materials and Methods” and “Results” section as follow:

“In a second experiment, the transcripts with the largest expression levels from RNA sequencing were confirmed using quantitative real-time polymerase chain reaction. The experiment was done in accordance with the procedure described above. Primer 5.0 was used to generate the oligonucleotides (S1 Table). Total RNA was extracted with a Qiagen RNeasy kit, and cDNA was prepared with oligo-dT primers and a SuperScript III kit (Invitrogen, Carlsbad, CA, USA). A Rotor-Gene Q PCR equipment (Hilden, Germany) and the QuantiTect SYBR Green PCR Kit were used for real-time quantitative PCR (Qiagen). During RT-qPCR gene quantification, the B. aryabhattai 16S rRNA and A. thaliana -actin genes were selected as internal controls. The real-time PCR reaction conditions were as follows: an initial denaturation step at 95 °C for 15 minutes, followed by denaturation at 95 °C for 15 seconds, an alignment step at 58 °C for 30 seconds, and an extension step at 72 °C for 40 cycles. Q-Gene software was used to calculate relative gene expression as mean normalized expression [35]. The relative fold-change (log2) values were calculated in relation to the control treatment. All quantitative PCR experiments were replicated three times for biological and technical considerations.”

“Even though RNA-Seq is the fundamental basis for gene expression profiling, qPCR is the main tool for validation. qRT-PCR analysis was done to validate the data generated by RNA sequencing. The expression levels of plant and bacterium genes matched the RNA-Seq data, indicating that the RNA-Seq results are trustworthy (Fig 7). Cinnamyl alcohol dehydrogenase (106-log2 ratio), apyrase (57-log2 ratio) and thioredoxin H8 (35- log2 ratio) were highest expressed genes in Arabidopsis plants inoculated with B. aryabhattai in a different replication of the experiment. While the zinc finger C-x8-C-x5-C-x3-H type family protein gene (-27-log2 ratio) was highest repressed during the interaction (Fig 7A). Besides, arginine decarboxylase (8-log2) and D-hydantoinase (6-log2) were highest expressed bacterium genes in planta. Meantime, the putative universal stress protein gene (-3-log2) had the most reduced expression (Fig 7B).”

---

## [Decision Letter · Decision Letter 1]

21 Jul 2022

Endophytic bacterium Bacillus aryabhattai induces novel transcriptomic changes to stimulate plant growth.

PONE-D-22-09489R1

Dear Dr. Borras-Hidalgo,

We’re pleased to inform you that your manuscript has been judged scientifically suitable for publication and will be formally accepted for publication once it meets all outstanding technical requirements. There is an additional comment from the review below.

"when G. chinensis (Keng) was firstly appeared, the full name of genus is required to provide."

Kind regards,

Ching-Hong Yang

Academic Editor

PLOS ONE

Additional Editor Comments (optional):

Reviewers' comments:

Reviewer's Responses to Questions

**Comments to the Author**

1. If the authors have adequately addressed your comments raised in a previous round of review and you feel that this manuscript is now acceptable for publication, you may indicate that here to bypass the “Comments to the Author” section, enter your conflict of interest statement in the “Confidential to Editor” section, and submit your "Accept" recommendation.

Reviewer #1: All comments have been addressed

Reviewer #2: All comments have been addressed

2. Is the manuscript technically sound, and do the data support the conclusions?

Reviewer #1: Yes

Reviewer #2: Yes

3. Has the statistical analysis been performed appropriately and rigorously? 

Reviewer #1: Yes

Reviewer #2: Yes

4. Have the authors made all data underlying the findings in their manuscript fully available?

Reviewer #1: Yes

Reviewer #2: Yes

5. Is the manuscript presented in an intelligible fashion and written in standard English?

Reviewer #1: Yes

Reviewer #2: Yes

6. Review Comments to the Author

Reviewer #1: (No Response)

Reviewer #2: When G. chinensis (Keng) was firstly appeared in the M&M, the full name of genus is required to provide.

7. PLOS authors have the option to publish the peer review history of their article (what does this mean?). If published, this will include your full peer review and any attached files.

Reviewer #1: **Yes: **Jian Huang

Reviewer #2: No

---

## [Editor Report · Acceptance letter]

25 Jul 2022

PONE-D-22-09489R1 

Endophytic bacterium *Bacillus aryabhattai* induces novel transcriptomic changes to stimulate plant growth. 

Dear Dr. Borras-Hidalgo:

I'm pleased to inform you that your manuscript has been deemed suitable for publication in PLOS ONE. Congratulations! Your manuscript is now with our production department. 

Kind regards, 

on behalf of

Dr. Ching-Hong Yang 

Academic Editor

PLOS ONE